# Optical Coherence Tomography in Inherited Macular Dystrophies: A Review

**DOI:** 10.3390/diagnostics14090878

**Published:** 2024-04-24

**Authors:** Alba Gómez-Benlloch, Xavier Garrell-Salat, Estefanía Cobos, Elena López, Anna Esteve-Garcia, Sergi Ruiz, Meritxell Vázquez, Laura Sararols, Marc Biarnés

**Affiliations:** 1Oftalmologia Mèdica i Quirúrgica (OMIQ) Research, c/Tamarit 39, 08205 Sabadell, Spain; xgarrell@omiq.es (X.G.-S.); elopez@omiq.es (E.L.); sruiz@omiq.es (S.R.); mvazquez@omiq.es (M.V.); lsararols@omiq.es (L.S.); mbiarnes@omiq.es (M.B.); 2Department of Ophthalmology, Hospital General de Granollers, Av Francesc Ribas s/n, 08402 Granollers, Spain; 3Hospital Universitari de Bellvitge, c/De la Feixa Llarga s/n, 08907 L’Hospitalet de Llobregat, Spain; estefaniacobos@hotmail.com; 4Clinical Genetics Unit, Laboratori Clinic Territorial Metropolitada Sud, Hospital Universitari de Bellvitge, Institut d’Investigació Biomèdica de Bellvitge (IDIBELL), c/De la Feixa Llarga s/n, 08907 L’Hospitalet de Llobregat, Spain; aesteveg@bellvitgehospital.ca

**Keywords:** macular dystrophy, OCT, tomography, biomarker, Stargardt, Sorsby, autosomal dominant drusen, occult macular dystrophy, pattern dystrophy, best vitelliform macular dystrophy

## Abstract

Macular dystrophies (MDs) constitute a collection of hereditary retina disorders leading to notable visual impairment, primarily due to progressive macular atrophy. These conditions are distinguished by bilateral and relatively symmetrical abnormalities in the macula that significantly impair central visual function. Recent strides in fundus imaging, especially optical coherence tomography (OCT), have enhanced our comprehension and diagnostic capabilities for MD. OCT enables the identification of neurosensory retinal disorganization patterns and the extent of damage to retinal pigment epithelium (RPE) and photoreceptor cells in the dystrophies before visible macular pathology appears on fundus examinations. It not only helps us in diagnostic retinal and choroidal pathologies but also guides us in monitoring the progression of, staging of, and response to treatment. In this review, we summarize the key findings on OCT in some of the most common MD.

## 1. Introduction

Macular dystrophies (MD) are a group of monogenic-inherited retinal disorders that commonly affect the vision of both eyes. The diagnosis, assessment, and ongoing monitoring of these conditions can pose significant challenges due to the extensive range of phenotypic variations, even within monogenic MD. The prognosis can also vary widely between different MDs. 

Optical coherence tomography (OCT) represents a non-invasive method for diagnosing conditions by providing a detailed cross-sectional view of the retina in real time. This technology relies on interferometry to construct a precise map of the retina’s layers with a high level of accuracy [1]. Early OCT systems operated in a time-domain manner; however, the newer technology known as spectral domain OCT (SD-OCT) and swept-source OCT (SW-OCT) are able to operate at increased speeds with enhanced sensitivity and signal-to-noise ratios, particularly at deeper locations beneath the RPE. Compared to SD-OCT, SW-OCT devices improve the visualization of structures beneath the RPE due to decreased sensitivity roll-off and attenuation of the OCT signal in deeper structures, particularly the choroid. To overcome this limitation in SD-OCT, techniques such as enhanced-depth imaging (EDI) are used to better visualize the choroid and structures below the RPE (see Table 1) [2]. 

Recent advances in fundus imaging, particularly OCT, have improved our understanding and the diagnosis of MD. Furthermore, OCT frequently identifies subtle morphological alterations that may go unnoticed during a fundus examination. The literature describes several OCT biomarkers associated with MD. Biomarkers, also known as biological markers, are characterized as medical indicators that signify either a diseased or non-diseased condition, serving as indicators for disease progression and prognostic.

In this narrative review, we performed a broad search of the literature on PubMed, initially including the most relevant studies using the following terms: “optical coherence tomography OR macular dystrophy” and “optical coherence tomography OR inherited retinal disease”. Additionally, we searched for the term “optical coherence tomography” in association (OR/AND) with each IRD described in the manuscript. Finally, we supplemented the search in some cases by adding references from relevant articles. All articles considered and cited were written in English.

In this review, we summarize the key OCT findings in some of the most common MDs. This may support the recognition, prognostic and progression of these disorders while, at the same time, pointing towards areas that need further research to fill gaps in our knowledge. Additionally, we highlight proposed OCT biomarkers in MD that may support the diagnosis or prognosis of these entities. 

## 2. Stargardt Disease

Stargardt disease 1 (STGD1, OMIM #248200) is an autosomal recessive MD caused by homozygous or compound heterozygous variants in the *ABCA4* gene (chromosome 1p22) and characterized by the atrophy of the retinal pigment epithelium (RPE) and the loss of photoreceptors. It is the most common macular inherited retinal disease (IRD). No large population-based studies have been conducted to determine its frequency, but its estimated prevalence ranges between 1:8.000 and 10.000. The product of the *ABCA4* gene is the ABCA4 protein, an ATP-binding cassette transporter located in the photoreceptor outer segments that transports N-retinyledene-phosphatidylethanolamine and phosphatidylethanolamine. Its dysfunction leads to increased accumulation of all-trans and 11-cis retinoids that build up A2E and other bisretinoids in the form of toxic lipofuscin within the RPE [1]. The antioxidant properties of RPE melanin may modulate disease severity since studies conducted in albino Abca4^−/−^ rodents suggest an earlier onset and more severe course of disease than in their pigmented counterparts [3,4]. Over 2000 confirmed or potentially disease-causing variants in the *ABCA4* gene have been documented as contributors to STGD1.

The typical presentation is a bilateral, symmetric, and slowly progressive visual acuity loss. The most common age of presentation is in the teenage years, but it can be diagnosed from childhood onwards. Late-onset STGD1 presents with foveal sparing and, given its older age at presentation, needs to be differentiated from age-related macular degeneration (AMD). STGD1 usually leads to legal blindness due to a bilateral loss of visual acuity to 20/200 or less [5]. Prognosis may be better in certain subgroups of patients, like those with the p.G1961E variant, with a normal full-field electroretinogram (ERG), or with presentation at older ages.

Phenotypically, STGD1 can present either as macular RPE atrophy with a beaten-bronze appearance or a bull’s eye pattern, with or without flecks (fundus flavimaculatus; Figure 1). Peripapillary sparing on fundus autofluorescence (FAF) is characteristic but not pathognomonic, and a dark choroid may be present on fluorescein angiography in up to 80% of cases due to blocking of choroidal fluorescence by increased intracellular accumulation of lipofuscin. Several classifications exist based on fundus appearance [6], FAF [7], and full-field ERG [8]. 

There are many findings on OCT in STGD1. In the early stages of the disease and even before changes are observed in the fundus or on FAF, Burke et al. reported a thickening of the external limiting membrane (ELM; Figure 2) [9]. The authors attributed this change to a response of the Müller cells to photoreceptor structural changes, but the migration and retraction of the inner segment of the photoreceptors have also been suggested. 

The hallmarks of STGD1 (and many other macular dystrophies) on OCT are areas of macular RPE, photoreceptor, and choriocapillaris loss (Figure 3). In 2017, the Consensus of Atrophy Meeting (CAM) study group proposed a set of criteria to define atrophy on OCT, which included the following: (i) a region of choroidal hypertransmission ≥ 250 µm; (ii) a zone of attenuation or disruption of the RPE ≥ 250 µm; (iii) overlying photoreceptor degeneration; and (iv) the absence of an RPE tear [10]. These findings were collectively called complete RPE and outer retina atrophy or cRORA. Incomplete forms, with smaller regions of RPE loss and choroidal hypertransmission (iRORA), and incomplete and complete forms of photoreceptor degeneration without RPE loss (iORA and cORA, respectively), were also proposed [10]. These definitions were developed for geographic atrophy (GA) secondary to AMD. Since GA is a subset of all cRORA, the use of this terminology has also been expanded to other degenerative diseases; for example, Doheny Imaging Reading Center investigators used it for defining atrophy on OCT in STGD1 in the ProgStar study [11]. Nonetheless, its adoption in other IRDs is still uncommon, particularly for i/cORA, which may not occur in all degenerative diseases.

Additionally, the area of atrophy in STGD1 progresses. Its growth approximates 0.10 mm/year [12], which seems inferior to that reported in other degenerative diseases like geographic atrophy (GA) secondary to AMD, at ~0.33 mm/year. This progression can also be followed by changes in total retinal thickness and volume or by outer retinal thickness thinning after segmentation. There is a correlation between central foveal thickness and visual acuity loss in STGD1, suggesting that it is the loss of the foveal outer retinal tissue, both photoreceptors and the RPE, that causes the drop in vision [13]. 

Other features may be present in patients with STGD1. Flecks, yellowish deposits in the outer retina resembling drusen, are seen as subretinal or intraretinal hyperreflective deposits. Different types of flecks have been described on OCT based on their location, extent, etc. (Figure 3), which may reflect a process of material accumulation and resorption in the photoreceptor layer or just different stages in the evolution of these lesions. 

A particular phenotype in STGD1 patients is the optical gap (Figure 4). This describes the subfoveal absence of the ellipsoid zone (EZ) with concomitant thinning of the nearby outer nuclear layer (ONL), which may eventually progress to a collapse of the inner retina on a thinned RPE. This finding has been associated with the p.G1961E variant in the mild end of the spectrum of STGD1 genetic variants [14]. Interestingly, this finding demonstrates photoreceptor loss preceding RPE atrophy. Similar findings are observed in some patients with other IRDs, like cone dystrophy and achromatopsia. 

Recently, Heath Jeffery et al. showed that patients with STGD1 have a thicker band 4 (corresponding to the RPE/Bruch’s membrane) than healthy controls and patients with *PRPH2* pattern dystrophies [11]. In addition, they proposed that the ratio of band 2 (corresponding to the EZ) to band 4 may help discriminate between these two IRDs, with a cutoff threshold of 0.79 providing 100% specificity [15]. Clearly, the study of individual layers on OCT (for integrity, thickness, or, recently, relative reflectivity in the case of the EZ [16]) is promising in this and other diseases. Nonetheless, the segmentation of outer retinal layers in STGD1 is difficult, and several limitations need to be taken into account when measuring some of these parameters.

Arriago et al. described four different choroidal patterns in STGD1 patients. More advanced patterns, with reduced Sattler and Haller layers ± choroidal caverns, significantly correlated with a more severe loss of retinal structural integrity, best corrected visual acuity (BCVA) loss, and different progression of the disease [17].

In addition, Piri et al. described the presence of choroidal hyperreflective foci in these patients (Figure 3), which presumably represent migrated lipofuscin granules into the choroid [18]. These foci were positively correlated with disease severity.

Optical coherence tomography can also be useful in the identification of the extent of foveal involvement. Short-wavelength light is absorbed by macular pigment. Therefore, on conventional (blue) FAF, both fovea and atrophy are hypoautofluorescent, making the detection of foveal atrophy difficult. This is important in eyes with foveal sparing, which is common in late-onset Stargardt. Foveal B-scans may unambiguously identify if the center of the foveal avascular zone is uninvolved or affected by iORA, cORA, iRORA, or cRORA. This may also be important in the setting of clinical trials, where the absence of foveal involvement may be one of the criteria for potential eligibility.

Structural OCT can detect subretinal and intraretinal fluid in the uncommon occurrence of macular neovascularization (MNV) associated with the disease, and it can also monitor the response to antiangiogenic treatment [19]. Optical coherence tomography angiography (OCTA) allows the visualization of the actual MNV in many IRDs, but its use in this situation in STGD1 is rare [19]. Nonetheless, OCTA has shown a decreased density of the superficial and deep capillary plexus, as well as the choriocapillaris, in STGD1 [20]. 

Finally, most studies of STGD1 determine disease progression by enlargement of areas of definitely decreased fundus autofluorescence (also known as DDAF), as reported in the ProgStar study [21]. However, the visualization of the OCT reflectance image using an *en face* projection (coronal view) and a subRPE slab reveals that atrophy is seen as a hyperreflective area and that there is a good correlation between areas of atrophy measured on FAF and *en face* OCT [22]. The STAR phase 2b trial (NCT03364153) evaluates the safety and efficacy of the C5 avacincaptad pegol (IVERIC bio, an Astellas company, Middlesex County, NJ, USA) as compared to sham in patients with STGD, and it is the only currently active trial that uses OCT-derived measurements as the primary endpoint (www.clinicaltrials.gov, accessed on 24 October 2023). In this case, the investigators evaluate the mean rate of change in the area of the EZ defect measured on *en face* OCT to determine disease progression. It is reasonable to assume that more researchers in the future will use OCT parameters as the primary endpoint for STGD1 trials.

## 3. Best Vitelliform Macular Dystrophy

Best vitelliform macular dystrophy (BVMD, OMIM #153700) is an IRD primarily caused by autosomal dominant variants in the *BEST1* gene (OMIM #607854) in 11q12. This gene encodes the integral membrane protein bestrophin-1, a calcium-activated chloride channel located on the basolateral membrane of the RPE cells [23]. The disruption of ionic transport caused by the mutation leads to the accumulation of fluid between the photoreceptors and the RPE, with a secondary accumulation of non-phagocytosed photoreceptor outer segments and their fluorophores in the form of lipofuscin in this space. An altered cholesterol homeostasis and RPE/photoreceptor apposition contribute to a weakened adhesion between the macular RPE and neurosensory retina. BVMD is the second most common inherited macular dystrophy and the most prevalent among autosomal dominant forms, with an estimated prevalence between 1.5 in 100,000 and 1 in 10,000 [24].

The classic form of BVMD is associated with hyperopia and typically emerges in the first two decades of life and leads to slow central vision loss, with high intra- and interfamilial variability. It is characterized by a distinctive oval lesion resembling an egg yolk (vitelliform lesion) in color fundus photographs. This lesion exhibits intense hyperautofluorescence [25]. On electrophysiological tests, full-field ERG results are generally normal, but electrooculogram (EOG) findings are often abnormal, with an Arden ratio of 1.5 or less. Genetic testing is typically used to confirm the diagnosis by identifying pathogenic *BEST1* variants. The prognosis is relatively good for the preservation of useful visual acuity in at least one eye, although there is substantial intrafamilial variability [26].

Since subretinal vitelliform lesions can exhibit diverse structural changes over a patient’s lifetime, Gass introduced a staging system that relies on ophthalmoscopic observations: stage I (subclinical), stage II (vitelliform), stage III (pseudohypopyon; Figure 5), stage IV (vitelliruptive), and stage V (atrophic/fibrotic). Progress in retinal imaging has delivered new insights into structural, vascular, and functional aspects.

OCT has played a crucial role in enhancing the clinical description of BVMD. It enables the visualization of various aspects, such as the vitelliform material, the disruption and atrophy of the outer retina layers, the presence of fibrotic nodules beneath the RPE, and hyperreflective points within the ONL [27].

In the subclinical phase, the sole noticeable change in OCT is a thickening of the interdigitation zone involving cones and the apical processes of the RPE, potentially resulting from impaired phagocytosis of discarded photoreceptor outer segments [27]. The characteristic egg-yolk lesion associated with the vitelliform stage presents as a dome-shaped hyperreflective structure within the subretinal space. In contrast, the pseudohypopyon and vitelliruptive stages demonstrate the reabsorption of the hyperreflective material, replaced by subretinal fluid, along with the elongation of outer segments. The final stage is marked by the atrophy of the outer retinal layers and the RPE, accompanied by fibrotic scarring [27].

Stages II, III, and IV tend to exhibit similar levels of visual acuity and retinal sensitivity [28]. Hence, extensive research has been devoted to the identification of OCT biomarkers capable of evaluating the severity of the disease and predicting progression. Several OCT biomarkers have been documented:The integrity of the EZ is the most closely linked factor to a decline in visual acuity. Its intensity, both in the affected areas and the surrounding regions, is attenuated in patients who show disease progression, and expansion of the disrupted EZ region is linked to a higher probability of visual acuity loss. Additionally, it is worth noting that many patients exhibit a central region with a preserved EZ, referred to as an optically preserved islet or OPI, which is associated with improved visual outcomes [29]. However, it is essential to recognize that EZ loss tends to occur in later stages and is typically associated with already significant visual impairment.Another noteworthy OCT biomarker is the measurement of ONL thickness, which can serve as an indicator of early photoreceptor degeneration. Irrespective of the clinical stage, BVMD patients exhibit reduced ONL thickness compared to healthy individuals [30]. Also, there is a correlation between ONL thickness and both BCVA and retinal sensitivity. Therefore, ONL thickness could prove to be a more valuable quantitative measure for assessing the anatomical loss of photoreceptors and the decline in macular function.Visual acuity is directly impacted by the presence of both vitelliform material and subretinal fluid. Eyes exhibiting vitelliform material typically demonstrate superior visual function, while the gradual reabsorption of this material, accompanied by the detection of subretinal fluid, is associated with a decline in visual acuity. Additionally, sensitivity, as measured by microperimetry, declines as vitelliform material is reabsorbed and subretinal fluid accumulates [31].Retinal hyperreflective foci (HRF) are punctiform lesions, typically smaller than 30 μm, primarily located in the outer nuclear and plexiform layers of the neuroretina. Battaglia Parodi et al. identified the presence of HRF in BVMD and noted an increase in their number as the disease progressed [32]. HRF may constitute accumulations of microglia or RPE cells detached from Bruch’s membrane (BM) and migrated into the subretinal space, eventually reaching the neuroretina [33]. Recent data obtained using polarization-sensitive OCT, which provides tissue-specific contrast, has lent support to the theory by demonstrating the presence of RPE cells within the vitelliform material and the neuroretina. Consequently, HRF appears to reflect the progression of the disease and may serve as a valuable biomarker for monitoring BVMD.Researchers have also explored biomarkers related to choroidal angioarchitecture in BVMD. One study examined changes in choroidal thickness across various BVMD stages, revealing that stages featuring vitelliform deposits showed a thicker choroid, while atrophic/fibrotic stages exhibited a thinner choroid compared to healthy controls. The underlying hypothesis was that the metabolic strain on the RPE and photoreceptor layers, due to the accumulation of vitelliform material, could lead to increased blood flow and the dilation of choroidal vessels, consequently resulting in greater choroidal thickness. Conversely, in later stages marked by outer retinal degeneration and reduced metabolic demands, choroidal thickness would be expected to decrease [34]. However, a more recent investigation observed no significant difference in subfoveal choroidal thickness between BVMD and control eyes [35]. On a different note, the choroidal vascularity index, a biomarker reflecting the relative ratio of vascular to stromal components in the choroid, exhibited a noteworthy decrease in BVMD irrespective of its stage [35].Additional OCT findings that may be detected in the later stages of BVMD include choroidal excavation and full-thickness macular holes. Choroidal excavation has been documented in individuals with BVMD and could be regarded as an indication of heightened degeneration in the outer retinal layers and the choroid [36]. Full-thickness macular holes, though rare, can be a potential complication of BVMD, as suggested by recent OCT-based research. Progressive retinal atrophy may be the most likely cause of the development of these holes.

Finally, the use of OCTA in clinical applications has enabled the identification of MNV without the necessity of intravenous dye (Figure 6). Recent investigations have revealed a significantly higher prevalence of MNV in BVMD, up to 65% of cases, which may be attributed to the utilization of OCTA. These lesions respond well to antiangiogenic therapy. Additionally, OCTA has unveiled a reduction in vessel density within the superficial capillary plexus, deep capillary plexus, and choriocapillaris in the macular region when compared to control subjects [35,37]. This finding underscores the role of the vascular system in the pathogenesis of BVMD.

## 4. Pattern Dystrophies

Pattern dystrophies (different OMIM numbers) are a spectrum of MDs characterized by the presence of a yellowish-grey deposit at the level of the RPE in the macula, each dystrophy presenting a characteristic distribution or pattern. There is an accumulation of lipofuscin at the level of the RPE, usually bilateral and symmetric. Pattern dystrophies can occasionally be associated with other systemic diseases, such as Steinert’s myotonic dystrophy, Kjellin’s syndrome, or pseudoxanthoma elasticum. In most cases, these conditions result from genetic variants within the *PRPH2* gene, located on chromosome 6p21.1, with an autosomal dominant inheritance pattern. This gene codes for the protein peripherin-2, a surface glycoprotein found in both cone and rod photoreceptor cell discs, which is involved in the adhesion and compaction of the discs. It is thought that the lack of the disulfide bond held by the amino acid cysteine, which is affected by many of the mutations involving *PRPH2*, weakens the disc membrane of the photoreceptors. Less frequently, other genes can cause pattern dystrophies (*BEST1*, *IMPG1*, *IMPG2,* and *CTNNA1*) [38].

Clinically, there are five types of pattern dystrophies based on the pigment distribution pattern: butterfly-shaped pigment dystrophy (BPD), adult-onset foveomacular vitelliform dystrophy (AOFVD), multifocal pattern dystrophy simulating STGD1, reticular dystrophy, and *fundus pulverulentus*. BPD is characterized by the presence of yellowish pigment deposits above the RPE, simulating butterfly wings with a radial distribution from the fovea, which can be readily identified on FAF (Figure 7). In AOFVD, we observe the formation of a bilateral and symmetrical round yellowish lipofuscin deposit under the neurosensory retina that is hyperautofluorescent on FAF and has a smaller size and later onset than in BVMD; although patients are generally asymptomatic, during their evolution, the deposit can regress and leave and RPE can atrophy, which initially causes metamorphopsia and, later, a loss of visual acuity. In the multifocal pattern dystrophy, there are scattered yellowish-white macular and perimacular deposits resembling flecks similar to those seen in STGD1 (but these are hyperautofluorescent and also have a rim of hypoautofluorescence on FAF), and there is no peripapillary sparing or silent choroid on fluorescein angiography characteristic of STGD1; eventually, macular atrophy can occur. Reticular dystrophy initially presents with a network of hyperpigmentation resembling a fishing net that gradually expands from the macula and can evolve into RPE atrophy; these changes are clearly seen in fluorescein angiography. The *fundus pulverulentus* is the least common dystrophy, with numerous irregular clumps of pigment conferring a punctate or granular appearance to the macula and the absence of drusen.

The earliest signs can be seen between the ages of 20 and 40, although most patients are asymptomatic. Despite their usually benign course, the development of RPE atrophy [39] or MNV can cause a significant visual disability in a minority of cases.

The OCT in pattern dystrophies shows the presence of a hyperreflective material between the EPR and the neurosensory retina throughout the different patterns. In BPD, areas of variable reflectivity can be observed between RPE/Bruch’s membrane and photoreceptors. In some cases, a disruption of the photoreceptors and occasionally hyperreflective bands with extension to the ONL can be seen [36]. In reticular dystrophy, we can observe multiple hyperreflective areas at the level of the RPE associated with focal losses of the photoreceptor outer segment, as well as of the EZ. In multifocal dystrophy, hyperreflective areas are between the RPE/Bruch’s membrane complex and the EZ, with focal disruptions in the outer segments. In some cases, the extension of the hyperreflectivity from the RPE to the ONL can be seen [40]. 

The use of OCT is important to differentiate AOFVD from AMD. AOFVD differs from the rest of pattern dystrophies in that it presents four stages throughout its evolution, as described with enhanced-depth imaging (EDI) [41]. In the first stage, a subretinal hyperreflective material is observed between the RPE and the dome-shaped and amputated EZ without significant visual acuity loss. The second stage is called the pseudohypopion phase and is similar to that seen in BVMD; intraretinal pseudocysts can be observed. In the vitelloruptive stage, the lesion collapses. The subretinal material gradually becomes more heterogeneous, and a loss of photoreceptors can be observed. In the last stage, complete reabsorption of the fluid is observed with a loss of photoreceptors and RPE atrophy. 

A high frequency of HRF has been observed in pattern dystrophies. In vitelliform lesions of different origins (not just pattern dystrophies), HRF corresponded to RPE cells with lipofuscin and melanolipofuscin granules, as well as melanosomes, and were associated with the development of outer retinal atrophy [33]. Casillo et al. [42] found that after the treatment of neovascular membrane, HRF persisted in pattern dystrophies; on the other hand, in AMD, the number of HRF decreased after the treatment. 

To differentiate ABCA-4- and PRPH2-associated retinopathy using OCT, Heath Jeffery et al. developed a novel imaging biomarker. This biomarker consists of the ratio between band 2 (EZ) and band 4 (RPE/Bruch membrane). The band 2/band 4 ratio was significantly greater in the PRPH2 group than in both the ABCA4 and healthy control groups, allowing the discrimination between these two IRDs [15].

In recent years, the role of the choroid in pattern dystrophies has been studied. Choroidal thickness is similar between controls and eyes with pattern dystrophy and significantly thicker than in an exudative AMD population 3 years after MNV treatment, suggesting different pathophysiological mechanisms in the two diseases and a potential role of this finding in the differential diagnosis between these entities [42]. The authors speculated that the increased thickness in patients with pattern dystrophies as compared to those with exudative AMD may be caused by the elimination of vitelliform material via the choroid, with thickening of the Haller layer, secondary compression of the choriocapillaris in some eyes with resulting ischemia and MNV development [42]. However, a thickened choroid in eyes with pattern dystrophies as compared with age-matched healthy controls would be expected in this scenario, which is not a consistent finding to date. 

In recent years, OCTA has allowed the visualization of MNV in pattern dystrophies. Type 1 and 3 MNV can be seen and responded to ranibizumab therapy in a retrospective study with stability of visual acuity [43], although spontaneous improvement can occur irrespective of treatment with antiangiogenic therapy.

## 5. Sorsby Fundus Dystrophy

Sorsby fundus dystrophy (SFD, OMIM #136900) is an autosomal dominant MD with complete penetrance caused by pathogenic variants in the *TIMP3* (tissue inhibitor of metalloproteinase 3) gene, located on chromosome 22q12.3 [44]. As a consequence, the *TIMP3*-mediated extracellular matrix turnover is altered, which leads to a thickening of BM that impairs the flow of nutrients to the photoreceptors and their degeneration. Also, since TIMP3 inhibits the vascular endothelial growth factor (VEGF), its mutation promotes angiogenesis. Due to the paucity of SFD cases, no large epidemiological studies exist that determine the prevalence of the disease in the general population, but its estimated prevalence is closer to 1 in 220.000 people.

The average onset of the disease is usually during the fourth to fifth decade of life. Clinical symptoms at presentation typically include poor dark adaptation (which may respond to vitamin A supplementation) [45], reduced color vision, central scotomas, metamorphopsia, and sudden loss of central vision [46]. Patients have worse scotopic than photopic visual function, as confirmed by low-luminance visual acuity, dark adaptometry, and scotopic microperimetry [47]. Long-term prognosis is generally poor because of the development of MNV, polypoidal choroidal neovascularization, fibrosis, and/or chorioretinal atrophy [22]. Sivaprasad et al. found that the median age for the occurrence of MNV in the first eye was 46 years and 50 years in the second eye, while the median age for legal blindness (visual acuity < 20/200) was 54 years [48]. 

The funduscopic findings may include the presence of drusen in the posterior pole and along the vascular arcades [46], mostly on the temporal side. Reticular pseudodrusen, deposits located between the photoreceptors and the RPE are also common in SFD. Although they can be seen as a yellowish deep network in the superior retina, they are most clearly observed as a reticular pattern on FAF (Figure 8) or as hyporeflective round lesions on near-infrared imaging. Patients show a characteristic macular late-phase hypocyanescence on indocyanine green angiography [49].

The main OCT findings are drusen, with reticular pseudodrusen appearing later [50] but before extensive chorioretinal atrophy ensues. They present a notable thickening of BM as the primary pathological change, as documented by Iyer et al. [50]. These alterations lead to the degeneration of the choriocapillaris and the outer retina. As a consequence, there is a thinning of the choriocapillaris in SFD patients [50]. Some studies suggest that a compromised choriocapillaris and Bruch’s membrane complex is linked to the degeneration of the photoreceptors above. Additionally, patients with SFD exhibit significant choroidal thinning as measured using horizontal enhanced-depth imaging [51]. 

Another characteristic finding in SFD is the presence of a separation in the RPE/Bruch’s membrane complex observed in OCT imaging. For patients with SFD, the extent of RPE/Bruch’s membrane separation was found to be correlated with the stage of the disease: those with the most severe ocular symptoms displayed the greatest degree of separation between the BM and RPE [52]. 

Patients with SFD can develop MNV at any point, either concurrently with drusen deposits or even in their absence. Active MNV causes intraretinal and subretinal fluid accumulation and subretinal hemorrhages. These findings can be detected through fundoscopic examination, OCT, and fluorescein angiography. The MNV responds to treatment with anti-VEGF injections [53]. 

OCTA has also been shown to identify early MNV without the need for fluorescein angiography [54]. The assessment of choroidal blood flow could also hold clinical significance for individuals with SFD who present reticular pseudodrusen. This is highlighted by recent research, which has shown that patients with AMD, reticular pseudodrusen, and impaired vision are linked to notably larger regions of reduced blood flow in the choriocapillaris on OCTA [55]. Prospective investigations employing OCTA imaging techniques have the potential to be valuable in identifying the early stages of the disease and predicting its progression, along with the likelihood of vision deterioration.

## 6. Autosomal Dominant Drusen 

Autosomal dominant drusen (ADD, OMIM #126600) encompasses both *Malattia Leventinese* and Doyne honeycomb retinal dystrophy. It is an autosomal dominant MD characterized by early-onset macular drusen-like deposits. It is caused by a single pathogenic variant (p.Arg345Trp) in the gene *EFEMP1*, located on chromosome 2p16.1, encoding the epidermal growth factor (EGF)-containing fibulin-like extracellular matrix protein 1 [56]. The precise function of the protein remains unknown.

Patients may have macular drusen-like deposits from the third decade and are usually asymptomatic or can complain of mild decreased visual acuity or metamorphopsia [57]. After the fifth decade, central vision may deteriorate due to extensive pigmentary changes, macular atrophy, or, rarely, MNV. Marked intra- and interfamilial variability is common [26]. 

Phenotypically, the main feature of ADD is the presence of round yellow-white or drusen-like deposits in the macula (Figure 9). These deposits become progressively confluent centrally, leading to a honeycomb appearance of the macula, often with RPE hyperplasia. Large drusen can also be characteristically found at the edge of the optic disc, while small drusen can be seen in the temporal macula in a radial pattern. The large drusen in ADD can be differentiated from typical drusen in AMD by their hyperautofluorescence on FAF. On both fluorescein angiography and indocyanine green, the large macular drusen turn from hypofluorescent in the early phases to hyperfluorescent in the late phases, while the reverse is true for small peripheral radial drusen [58]. 

Spectral-domain optical coherence tomography (SD-OCT) imaging can reveal focal dome-shaped, saw-toothed, or diffuse hyperreflective deposits with an elevation between the RPE and Bruch’s membrane, usually becoming more confluent over time (Figure 10) [59]. When focal, these drusen are similar to those seen in AMD, although some adopt a more nodular appearance. Small radial drusen range from a small thickening of the RPE/Bruch’s membrane complex to a sawtooth RPE elevation reminiscent of cuticular drusen [59]. Peripapillary drusen show similar findings. Zweifel et al. reported the presence of subretinal deposits on SD-OCT similar to reticular pseudodrusen in two members of the same family with an otherwise ADD-like phenotype, but in whom variants in *EFEMP1* gene could not be detected [60]. 

Functional impairment, as evaluated by microperimetry, correlates topographically to the sub-RPE deposition of drusenoid material [59]. Therefore, the greater the hyperreflective material deposited below the RPE, the greater the visual impairment.

As with Sorsby disease, a separation between the RPE and Bruch’s membrane has also been described, which increases as the stage of the disease progresses. On the contrary, this relationship has not been found in late-onset retinal degeneration [52].

A peculiar outer retina morphologic change, occurring in a variety of advanced degenerative retinal disorders, termed outer retina tubulation [59] has also been detected. It is a feature of photoreceptor rearrangement after retinal injury.

Early in the disease, the outer retina (photoreceptors) may remain intact, but later stages can show variable or diffuse ellipsoid zone loss, as well as outer and inner segment disruption. Querques et al. described that the inner segment (IS)/outer segment (OS) junction of the photoreceptors overlying the focal dome-shaped or diffuse RPE elevation is disrupted; however, the IS/OS junction overlying the sawtooth RPE elevation is normal (Figure 10) [59].

Drusen may evolve into areas of RPE atrophy and, rarely, MNV. These complications can already be detected in OCT and OCTA [61], respectively. 

## 7. Occult Macular Dystrophy

Occult macular dystrophy (OMD, OMIM #613587), also called Miyake disease, is an autosomal dominant MD characterized by a progressive loss of visual acuity without ophthalmoscopic or angiographic changes. Although, in most cases, no genetic cause is found, several variants within the *RP1L1* gene, located on chromosome 8p23, have been described [62]. 

As studied by the East Asia Occult Macular Dystrophy study group, patients usually report a loss of visual acuity (in the range from 20/16 to 20/320, with a median of 20/100), which usually begins around 25 years (range, 2 to 73) [63]. The diagnosis is made with electroretinographic tests since both FAF and fluorescein angiography are eminently normal. Full-field ERG usually does not show abnormalities, while focal ERG does show a depression of macular function, confining the disease to the macular area. With the development of the multifocal ERG, it has been possible to detect a reduction in the amplitudes of the central areas. Occasionally, patients may not present a decrease in vision despite having a reduction in the focal ERG. Visual acuity may worsen over 10–15 years and stabilize thereafter, although high variability between patients is the norm [63]. 

Careful attention to foveal OCT scans shows subtle abnormalities despite the normal fundus appearance. The classical OCT findings include the blurring of the EZ and the absence of interdigitation of the photoreceptors with normal RPE [63]. Nakamura et al. published a large study where they proposed a three-stage classification [64]. In stage 1, patients are asymptomatic, and most show minimal alterations on OCT, although effacement of the EZ and the interdigitation zone may be observed. In a study by Kato et al. with 40 patients, it is shown that in asymptomatic patients with an *RP1L1* genetic variant, the integrity of the foveal structures is maintained, probably representing early phases of the disease [65]. Then, in stage 2, there is a loss of the outer segment–RPE interdigitation zone, while the EZ is blurred and acquires a dome shape. Finally, in stage 3 the EZ looks flat and, in advanced stages, discontinuous. To calculate this flattening of the ellipsoid layer, Chen et al. developed a formula called *effective foveal outer segment length* (eFOSL), which consists of a subtraction between the distance between the ellipsoid layer and subfoveal interdigitation minus the distance between the ellipsoid layer and the interdigitation at three nasal degrees of the fovea [66]. In this study, they demonstrated that affected people have a lower eFOSL than normal controls, thus quantifying the absence of the bowing of the EZ [66]. Similar to this formula, Nakamura et al. measured the distance between the external limiting layer and the RPE (ERT), showing that visual acuity decreased as the ERT decreased [64]. Central retinal thickness is maintained during the early stages, with thinning in the late stages [64]. This thinning of the fovea correlates with loss of visual acuity, as it does in STGD1 and retinitis pigmentosa. It is interesting that the thickness of the interdigitation layer has been correlated with the foveal amplitude in the multifocal ERG, indicating that OCT findings explain the functional and electroretinographic results. 

## 8. Conclusions

OCT is an important tool in the diagnosis and management of patients with IRD (see Table 2). It evaluates the damage of the macular RPE and photoreceptors, which characterizes MD. It is also important for the identification of accompanying lesions like drusen and their subtypes, flecks, vitelliform lesions, and the integrity of outer retinal bands. On the other hand, OCTA is important for diagnosing MNV non-invasively, which is a frequent and treatable complication of some MDs like BVMD and SFD.

However, more research is needed to improve the differential and early diagnosis of MD to identify biomarkers of progression and responses to treatment. Imaging methods play an important role in all these areas. 

For example, OCT can support genotype–phenotype correlations (such as those observed for the p.G1961E variant in STGD1), drusen characterization in ADD or SFD to determine if they are different from those more commonly observed in AMD (and thus, if OCT can support the clinical differential diagnosis between these entities), or additional features in OMD to minimize misdiagnosis and unnecessary ancillary testing. 

We are aware that our study has several limitations. First of all, the availability of relevant studies and data on inherited macular dystrophies may be limited, leading to challenges in obtaining a comprehensive overview. Additionally, the understanding of inherited macular dystrophies and OCT imaging features may have evolved over time. Older studies may not reflect the current state of knowledge, and the interpretation of OCT images may have changed with advancements in technology and clinical experience. The field of OCT is rapidly evolving, with continuous technological advancements. If the review does not include the most recent studies or technologies, it may not capture the latest developments in the OCT imaging of macular dystrophies. Also, studies included in the review may vary in terms of sample size and the characteristics of the study populations. Small sample sizes may limit the statistical power of the analysis, and differences in demographics could affect the generalizability of findings. Moreover, inherited macular dystrophies encompass a diverse group of disorders with different genetic mutations, clinical presentations, and disease progressions. This heterogeneity can make it challenging to draw generalizable conclusions, and results may not apply uniformly to all forms of the condition. Lastly, it is a literature review, and the quality of individual studies included in the review can impact the overall reliability of the findings. Issues such as study design, methodology, and potential biases in data collection and analysis should be considered when interpreting the results.

Looking ahead, OCT will likely be used for the identification of novel biomarkers of progression and response to treatment. Given its resolution and ability to provide 3D visualization, the (semi)automatic quantification of topographic changes in RPE and EZ thickness and reflectivity is a potential avenue of development. This may benefit from *en face* visualization strategies and color-coded maps, which will require the correction of artifacts and proper automatic segmentation (possibly through the application of artificial intelligence algorithms.

In summary, OCT plays, and will likely play, a pivotal role in the diagnosis and follow-up of patients with MD. Multimodal imaging and comprehensive functional testing will contribute to a deep phenotypic characterization of these entities for the benefit of patients.

## Figures and Tables

**Figure 1 diagnostics-14-00878-f001:**
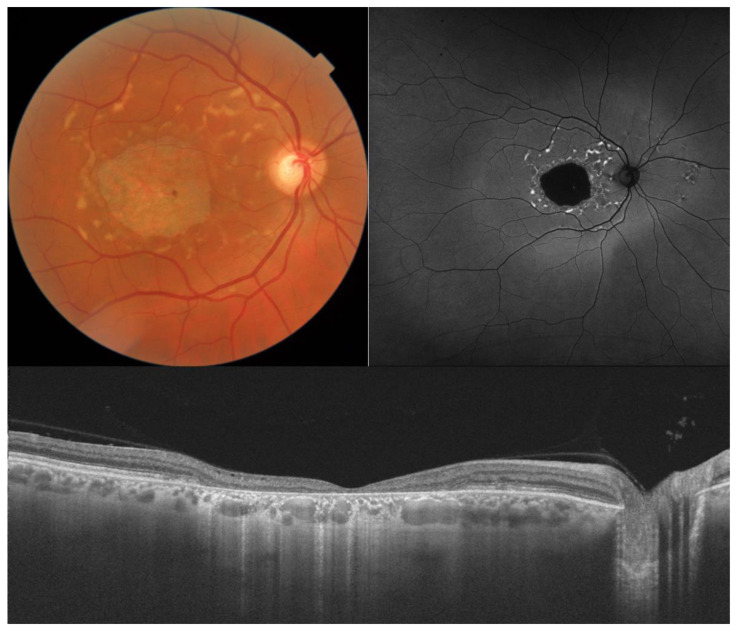
Color fundus photography, FAF, and OCT of a 58-year-old male with STGD1 and a visual acuity of counting fingers. Macular atrophy and flecks, which are clearly visible on FAF as a round area of hypoautofluorescence and bright, hyperautofluorescent discrete lesions, respectively. On OCT, there is a complete loss of retinal pigment epithelium, ellipsoid, external limiting membrane, and outer nuclear layer with secondary increased choroidal signal. The retina is markedly thinned, and the inner nuclear layer rests on Bruch’s membrane. FAF: fundus autofluorescence.

**Figure 2 diagnostics-14-00878-f002:**
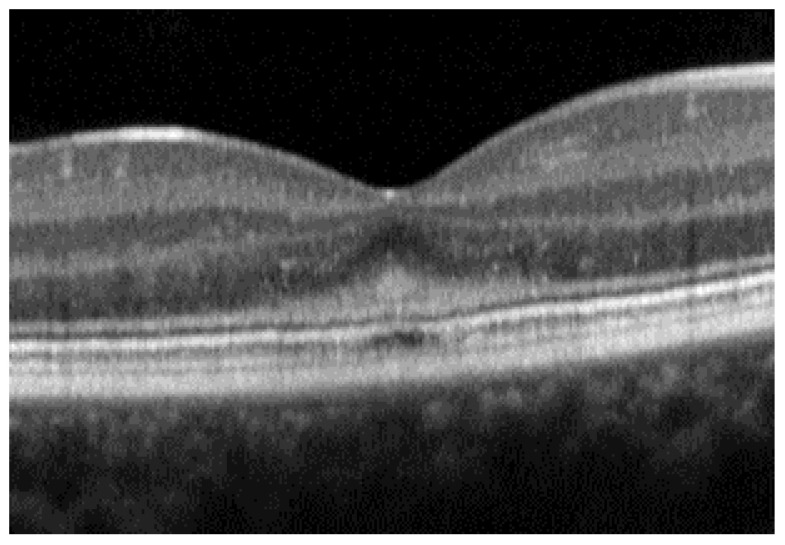
Thickening of the external limiting membrane in a 12-year-old female. Visual acuity was 20/20, and no abnormalities were found on clinical exam, color fundus photography, or fundus autofluorescence (not shown). The thickening is most noticeable under the foveal depression. The patient had pathogenic biallelic variants in the *ABCA4* gene.

**Figure 3 diagnostics-14-00878-f003:**
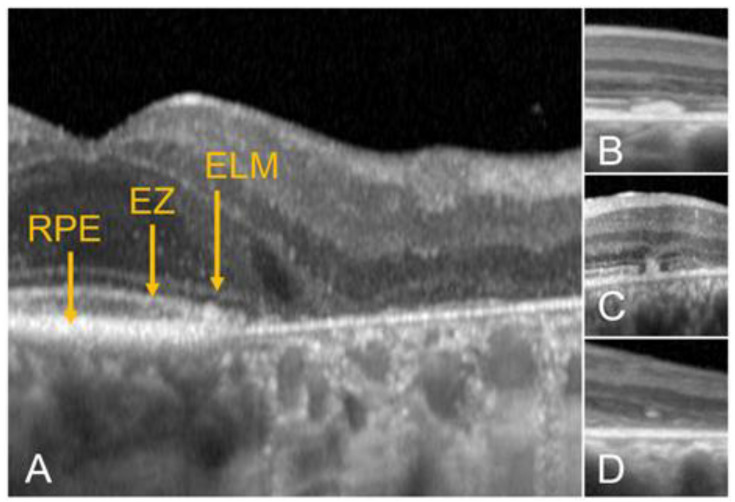
Areas of atrophy of the outer retina in STGD1 as seen on optical coherence tomography. Note hyperreflective foci in the choroid in (**A**). (**B**), subretinal fleck with loss of the ellipsoid zone. (**C**), a different fleck with breaks in the external limiting membrane. (**D**), migrated fleck into the retina.

**Figure 4 diagnostics-14-00878-f004:**
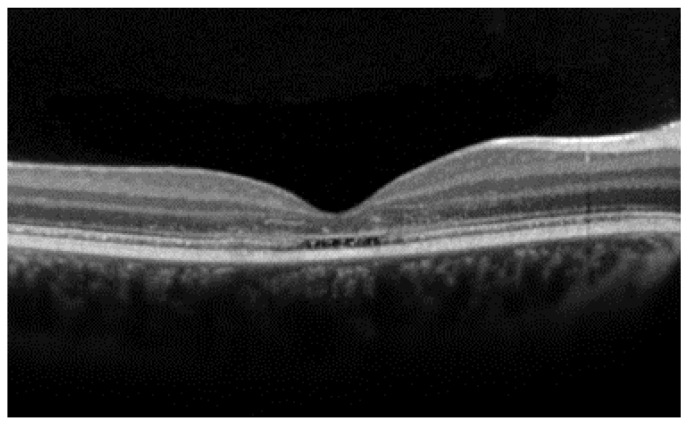
Optical gap in a 25-year-old female with 20/80 visual acuity. Loss of the subfoveal ellipsoid zone with relative preservation of the foveal thickness.

**Figure 5 diagnostics-14-00878-f005:**
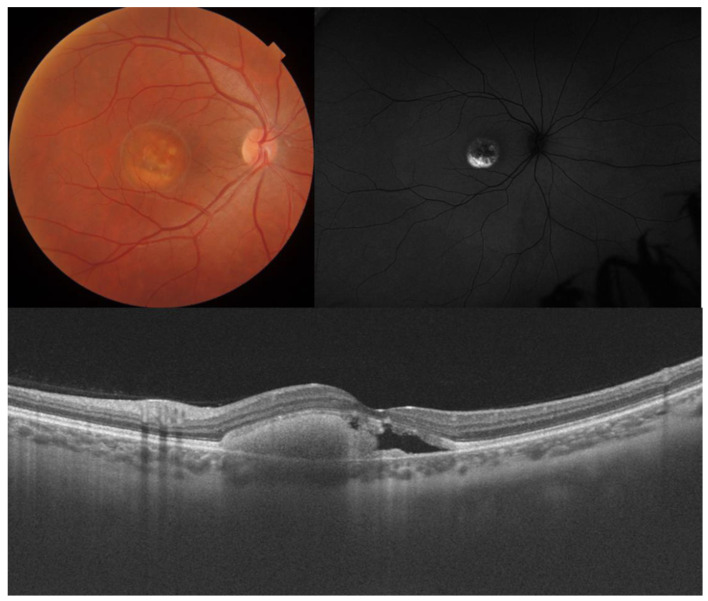
Color fundus photography, fundus autofluorescence, and OCT in a 24-year-old female with Best vitelliform macular dystrophy in the pseudohypopion stage and a surprisingly good visual acuity of 20/25. Notice that the inferior yellowish deposit (**top left**) is hyperautofluorescent on fundus autofluorescence (**top right**). (**Bottom**), a vertical OCT shows that the yellowish deposit is located in the subretinal space inferior to the fovea (to the **left** in the vertical B-scan), while the upper part of the lesion (to the **right**) is optically hyporreflective and can be confused with subretinal fluid from active macular neovascularization.

**Figure 6 diagnostics-14-00878-f006:**
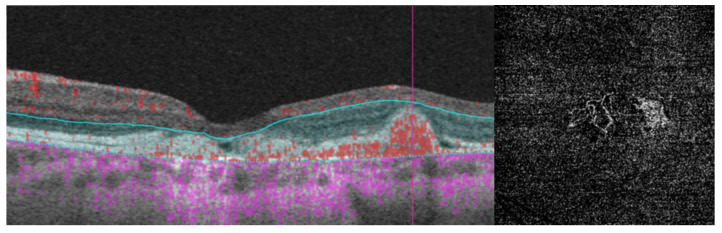
MNV in a female patient aged 22 years old with a visual acuity of 20/100 and Best vitelliform macular dystrophy as seen on OCTA. (**Left**), the angio B-scan shows a type 1 MNV between the retinal pigment epithelium and Bruch’s membrane. (**Right**), the *en face* image of the outer retina shows the actual vessels. MNV: macular neovascularization.

**Figure 7 diagnostics-14-00878-f007:**
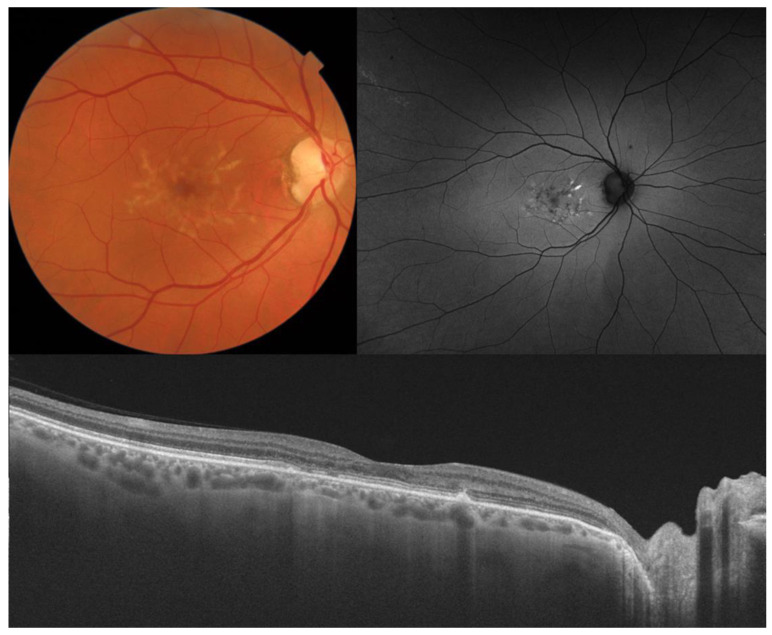
Color fundus photography, fundus autofluorescence, and OCT in butterfly-shaped pattern dystrophy. This 50-year-old male had 20/25 visual acuity. Notice the abnormalities in the retinal pigment epithelium and the ellipsoid zone, with absence of fluid and preservation of retinal thickness.

**Figure 8 diagnostics-14-00878-f008:**
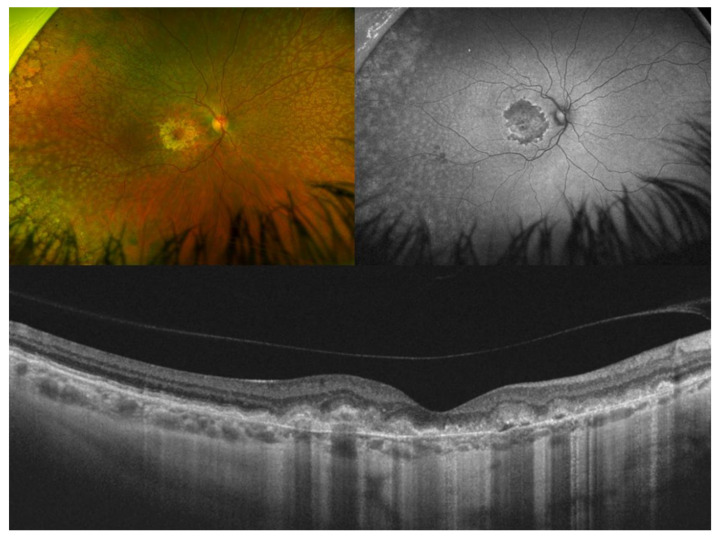
Color fundus photography, fundus autofluorescence, OCT in a male patient aged 55 years old, and visual acuity of 20/400 with Sorsby fundus dystrophy and variants in *TIMP3*. Notice the presence of extensive drusen and macular atrophy, with subretinal debris on OCT.

**Figure 9 diagnostics-14-00878-f009:**
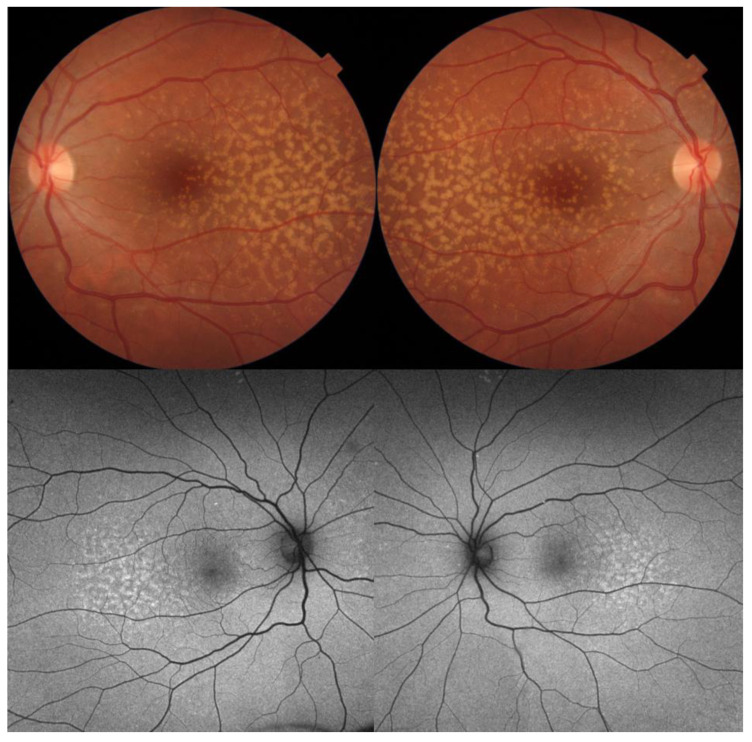
Color fundus photography and fundus autofluorescence in a woman genetically confirmed with autosomal dominant drusen aged 37 years old and 20/20 vision. There is a bilateral, temporal, and highly symmetrical distribution of the drusenoid deposits, which are hyperautofluorescent. In this case, there are no drusen adjacent to the optic disk.

**Figure 10 diagnostics-14-00878-f010:**
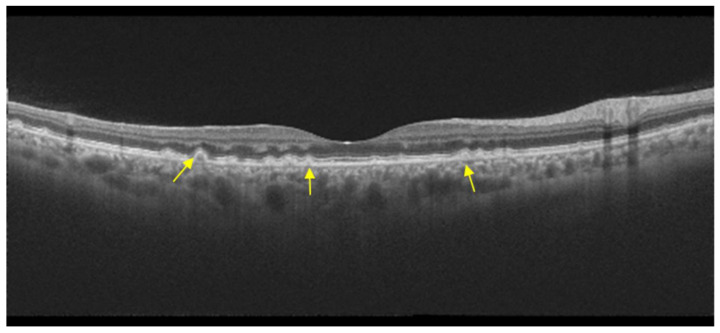
Optical coherence tomography in autosomal dominant drusen, with multiple drusen on the inner aspect of Bruch’s membrane (yellow arrows) and preserved IS/OS junction.

**Table 1 diagnostics-14-00878-t001:** Comparison of salient features of the three types of optical coherence tomography.

Type of OCT	Image Acquisition	Scanning Speed	Axial Resolution	Transverse Resolution	Range of Imaging
Time domain	Superluminescent diode (810 nm) single photon detector, moving mirror	400 A-scans per second	10 µm	20 µm	Vitreoretinal interface to RPE
Spectral domain	Broadband superluminescent diode source (840 nm), array of detectors, fixed mirror	27,000–70,000 A-scans per second	5–7 µm	14–20 µm	Posterior cortical vitreous to sclera using EDI mode
Swept source	Swept-source tunable laser (1050 nm), single detector	100,000–400,000 A-scans per second	5 µm	20 µm	Posterior cortical vitreous to sclera using EDI mode

RPE: retinal pigment epithelium, EDI: enhanced-depth imaging, OCT: optical coherence tomography. Comparison of salient features of the three types of optical coherence tomography from [2].

**Table 2 diagnostics-14-00878-t002:** Characteristics of optical coherence tomography in hereditary macular dystrophies.

Stargardt disease	Thickening of the ELM Thickening of the RPE/BM complex Optical gap: variant p.G1961E Flecks: hyperreflective material subRPE or intraretinal Atrophy: *en face* OCT can measure the progression of atrophy Reduced central macular thickness: the lower the thickness, the lower the BCVA. Different choroidal patterns: reduction in thickness of the Sattler AND Heller layer worse BVCA Choroidal and retinal hyperreflective foci: the greater the number, the greater the severity of the disease.
Best vitelliform macular dystrophy	EZ integrity: if there is attenuation of the EZ layer, greater progression and lower BCVA The OPI is a region of the preserved EZ layer associated with better BCVA Thickness ONL: less thickness, more advanced disease Presence of SRF: the replacement of vitelliform material by SRF is associated with scotomas in microperimetry Hyperreflective foci: the number increases as the disease progresses Reduced choroidal thickness Foveal choroidal excavation
Pattern dystrophies	Hyperreflective foci Preserved choroidal thickness Band 2/band 4 ratio increased
Sorsby fundus dystrophy	Reticular pseudodrusen BM thickening Thinning of the choriocapillaris Decrease choroidal thickness Separation between RPE/BM: the greater the separation, the more advanced the disease
Autosomal dominant drusen	SubRPE hyperreflective material Separation between RPE/BM: the greater the separation, the more advanced the disease External tubulations Altered EZ layer is associated with worse BCVA
Occult macular dystrophy	EZ erase Interdigitation layer erasure EZ flattening Reduced effective length of the foveal outer segment ERT: distance between the ELM and RPE layer. The shorter the distance, the lower the BCVA Reduced central macular thickness

ELM: external limiting membrane; RPE: retinal pigment epithelium; BM: Bruch’s membrane; BCVA: Best corrected visual acuity; EZ: ellipsoid zone; ONL: outer nuclear layer; SRF: subretinal fluid.

## Data Availability

Not applicable.

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
