# Peer review of "Optical Coherence Tomography in Inherited Macular Dystrophies: A Review"

_diagnostics, 2024, doi:10.3390/diagnostics14090878_

Round 1

Reviewer 1 Report

Comments and Suggestions for Authors

The authors have presented a review work on the key findings of OCT-base biomarkers for identifying the most common Macular dystrophies (MD). This is an interesting review paper. However, authors are advised to revise the paper based on the following important and minor comments.

1. Authors need to indicate the scale bars in all retinal images (Both fundus as well as OCT B-scans)

2. Authors should mention the expansion of all abbreviations when they are used first time. Eg: ELM (External limiting membrane)

3. I could not see the color fundus photograph in Fig 2. It is mentioned in Fig caption

4.  Line 88-90 : The melanin pigmentary changes are also recently identified as a hallmark of the STGD1 caused by ABCA4 deficiency, especially in the study performed on pre-clinical models [1-3]. Authors need to mention this important point by adding the suggested most relevant papers

(1) https://doi.org/10.1016/j.omtn.2022.04.015  (2) https://doi.org/10.1109/JSTQE.2016.2556226 

Author Response

The authors have presented a review work on the key findings of OCT-base biomarkers for identifying the most common Macular dystrophies (MD). This is an interesting review paper. However, authors are advised to revise the paper based on the following important and minor comments.

  1. Authors need to indicate the scale bars in all retinal images (Both fundus as well as OCT B-scans)

Thank you for your comments. In ophthalmology we do not usually use scale bars. All images were magnified and cropped as needed to highlight relevant details, and proportions were maintained in all of them. We are not aware of explicit scale bars in color fundus photography (CFP) or fundus autofluorescence (FAF) images. In fact, CFP and FAF images are a 2D representation of the retina (that is a 3D organ), that’s why the same machine does a correlation between measures and size in central retina and in periphery; for OCT, all images were captured with either the Topcon Triton® SSOCT or the Heidelberg Engineering Spectralis® HR+OCT in the high-resolution mode (digital image size 1536x1536) using 12 mm scan.

  1. Authors should mention the expansion of all abbreviations when they are used for the first time. Eg: ELM (External limiting membrane)

Thank you so much for your advice. We have reviewed all the abbreviations and their expansion. The abbreviation of ELM (limiting membrane) is on line 103. The expansion of RPE was added on line 63. We also added the expansion of BCVA on line 172. And finally, Bruch membrane (BM), on line 301.

3. I could not see the color fundus photograph in Fig 2. It is mentioned in Fig caption.

Thank you for the comment. The images were not shown, we have clarified this issue in the Fig caption (“Figure 2. Thickening of the external limiting membrane in a 12-year-old female. Visual acuity was 20/20 and no abnormalities were found on clinical exam, color fundus photography or fundus autofluorescence (not shown). The thickening is most noticeable under the foveal depression. The patient had pathogenic biallelic variants in the ABCA4 gene.”).

  1. Line 88-90 : The melanin pigmentary changes are also recently identified as a hallmark of the STGD1 caused by ABCA4 deficiency, especially in the study performed on pre-clinical models [1-3]. Authors need to mention this important point by adding the suggested most relevant papers

(1)https://doi.org/10.1016/j.omtn.2022.04.015  (2) https://doi.org/10.1109/JSTQE.2016.2556226 

We thank the reviewers for this issue. We have included this information and the corresponding references as follows: “The antioxidant properties of RPE melanin may modulate disease severity, since studies conducted in albino Abca4-/- rodents suggest an earlier onset and more severe course of disease than in their pigmented counterparts [3,4]” (Lines 71-3).

Reviewer 2 Report

Comments and Suggestions for Authors

The authors make a review about using OCT in diagnosis and follow up of inherited macular dystrophies and I think their work is comprehensive and very informative and useful for most of ophthalmologists to guide them for easily diagnosing of these cases. 

Author Response

The authors make a review about using OCT in diagnosis and follow up of inherited macular dystrophies and I think their work is comprehensive and very informative and useful for most of ophthalmologists to guide them for easily diagnosing of these cases. 

Thank you very much for your kind words. We are very grateful that this manuscript has been of interest to you.

Reviewer 3 Report

Comments and Suggestions for Authors

I would like to congratulate the authors on their work. A few issues that could improve their work are:

1.Please give a brief background on OCT since this is the main focus of your work. What are the main types of OCT techniques and devices used(e.g. SD-OCT, SS-OCT).

2.Please add a brief section on your search strategy since this is a review paper. Please include which databases were searched from the authors to arrive at the studies included in this review.

3.Please in order to make this article more easy to read, add a few tables. For instance add a table for Stargard's disease with the different characteristics of the condition and how they appear on OCT.
